

# An artificial neural network classification method employing longitudinally monitored immune biomarkers to predict the clinical outcome of critically ill COVID-19 patients

Gustavo Martinez[1,2,*], Alexis Garduno[3,*], Abdullah Mahmud-Al-Rafat[2], Ali Toloue Ostadgavahi[2], Ann Avery[4], Scheila de Avila e Silva[5], Rachael Cusack[3], Cheryl Cameron[6], Mark Cameron[7], Ignacio Martin-Loeches[3] and David Kelvin[1,2]

[1] Immunology, Shantou University, Shantou, GD, China
[2] Microbiology and Immunology, Dalhousie University, Halifax, Nova Scotia, Canada
[3] Department of Clinical Medicine, University of Dublin, Trinity College, Dublin, Ireland
[4] Division of Infectious Diseases, MetroHealth Medical Center, Cleveland, OH, United States of America
[5] Department of Biotechnology, Universidade de Caxias do Sul, Caxias do Sul, Rio Grande do Sul, Brazil
[6] Department of Nutrition, Case Western Reserve University, Cleveland, OH, United States of America
[7] Department of Population & Quantitative Health Sciences, Case Western Reserve University, Cleveland, OH, United States of America
[*] These authors contributed equally to this work.

Corresponding authors
Gustavo Martinez, gsmartinez@ucs.br
David Kelvin, david.kelvin@dal.ca

## ABSTRACT

**Background**. The severe form of COVID-19 can cause a dysregulated host immune syndrome that might lead patients to death. To understand the underlying immune mechanisms that contribute to COVID-19 disease we have examined 28 different biomarkers in two cohorts of COVID-19 patients, aiming to systematically capture, quantify, and algorithmize how immune signals might be associated to the clinical outcome of COVID-19 patients.

**Methods**. The longitudinal concentration of 28 biomarkers of 95 COVID-19 patients was measured. We performed a dimensionality reduction analysis to determine meaningful biomarkers for explaining the data variability. The biomarkers were used as input of artificial neural network, random forest, classification and regression trees, k-nearest neighbors and support vector machines. Two different clinical cohorts were used to grant validity to the findings.

**Results**. We benchmarked the classification capacity of two COVID-19 clinicals studies with different models and found that artificial neural networks was the best classifier. From it, we could employ different sets of biomarkers to predict the clinical outcome of COVID-19 patients. First, all the biomarkers available yielded a satisfactory classification. Next, we assessed the prediction capacity of each protein separated. With a reduced set of biomarkers, our model presented 94% accuracy, 96.6% precision, 91.6% recall, and 95% of specificity upon the testing data. We used the same model to predict 83% and 87% (recovered and deceased) of unseen data, granting validity to the results obtained.

**Conclusions**. In this work, using state-of-the-art computational techniques, we systematically identified an optimal set of biomarkers that are related to a prediction

capacity of COVID-19 patients. The screening of such biomarkers might assist in understanding the underlying immune response towards inflammatory diseases.

## INTRODUCTION

By the end of 2019, a respiratory virus started to alert health authorities by inflicting its victims with a severe acute respiratory syndrome, namely coronavirus disease (COVID-19). In little time, a global pandemic was declared by the World Health Organization (*He, Deng & Li, 2020*), which has caused health authorities and scientists to apply unmeasurable efforts to understand the intricacies of the novel disease. The virus has caused millions of deaths, hospitalizations, and infections at alarming rates. The harsh effects caused by the virus still pose a threat even more than two years after the initial onset of the pandemic. Factors such as the novelty of the virus, its high potential for transmission, and mutations find a considerable share of humankind prone to be infected.

The host response to diseases such as COVID-19 generates immune signals that might be used to explain or predict the severity of the disease (*Liu & Hill, 2020*; *Yang et al., 2020*). A group of cytokines, chemokines and other biomarkers was previously reported (*Bermejo-Martin et al., 2020*) as good indicators of the unbalanced host immune response caused by inflammatory diseases such as COVID-19. For instance, tumor necrosis factor (TNF) is a cytokine that is involved several cell signalling events, being a major regulator of inflammatory responses (*Jang et al., 2021*). Next, CCL2 is a chemokine that is actively involved in immune processes, promoting the recruitment of immune cells to the inflammatory site. Moreover, IL10 may play a central role in regulating cytokine storms, given that this protein has anti-inflammatory properties (*Iyer & Cheng, 2012*). Also, the biomarkers MPO, SPD, ICAM, LIPO, VCAM, GMCSF, and VEGFC are all markers of vascular tissue damage (*Zhao et al., 2014*; *Kong et al., 2018*), which might also indicate viral escape in the bloodstream as a result of severe manifestations of COVID-19 (*Bermejo-Martin et al., 2020*).

However, promoting the evaluation of certain immune proteins for predicting the clinical outcome of a patient is not as straightforward as it may seem; in fact, inflammatory patterns are very rapid and may change over time, increasing quickly in the first stages of infection and decreasing during the recovery stage (*Wang et al., 2020*). Furthermore, many cytokines/chemokines belong to complex pathways of interactions such as the interleukin 6 trans-signaling reported by (*Scheller et al., 2011*), which requires the analysis of each biomarker together with its interactants, requiring complex mathematical approaches to link biomarkers with the clinical pathway of patients.

Therefore, an extensive calculation approach is required so it can encompass the non-linear pathway of inflammatory markers in COVID-19. An example of a tool that has successfully been employed in assisting clinical decision making is artificial neural

networks (ANNs) (*Shahid, Rappon & Berta, 2019*). The ability to classify data found in these algorithms enables that the weighted sum of n given inputs be explored to derive a classificatory path between pre-defined classes (*Russell & Norvig, 2021*). The mathematical robustness of such a method has succeeded in assisting clinical decision making in a plethora of areas, including health informatics (*Andreu-Perez et al., 2015*; *Ravi et al., 2017*).

Standalone analyses of biomarkers' concentration have been identified as good predictors of severity as they succeed in capturing a dysbalanced host immune response. For example, sepsis and septic shock syndromes may cause hypoxia due to tissue hypoperfusion; then a transcription factor protein (*i.e.*, hypoxia inducible factor 1-alpha) might trigger immune cells, which tend to upregulate the expression of PDL1 and VEGFC (*Cao et al., 2009*; *Noman et al., 2014*). Moreover, high serum levels of TNF-a were observed as a good predictor of fatality in critically ill sepsis patients (*Yousef & Suliman, 2013*). Additionally, high levels of ICAM1 were found to be a good predictor of severity in sepsis from bacterial and viral sources, as the biomarker is upregulated in endothelial injury (*Kaur etal , 2021*). Notwithstanding, the use of multiple biomarkers has not yet been employed as characterizers of severity of disease as the addition of novel input variables scalarly contribute to more complex statistical analysis, rendering mechanistic statistics as non-viable approaches and potentially requiring the aid of machine learning to achieve better predictability (*Martinez et al., 2022*).

In the present study, we hypothesize that the non-linear relationship of a consortium of immune biomarkers can be used to represent an unbalanced immune response among COVID-19 patients. We aim to obtain an optimal set of biomarkers to serve as input to classificatory models, providing an inexpensive and fast *in-silico* model for selecting proteins that play a key role in explaining the inflammatory pattern of COVID-19 patients.

## MATERIALS & METHODS

### Train/test dataset

One clinical cohort of COVID-19 ICU patients was considered for training/testing the model. This study took place in Cleveland, USA. Patients were enrolled from March 2020 to May 2020. Patients were hospitalized either because they were potential candidates for mechanical ventilation and/or because they were judged to be in an unstable condition requiring intensive medical or nursing care. This cohort was composed of 45 patients. The patients had longitudinal samplings on their biomarkers consisting of multiple time points (first day of admission, fifth day, eighth day, eleventh day, and 15th day) unevenly spread. All the samples associated with each patient were averaged and associated with a clinical outcome, *i.e.*, deceased or recovered. A panel of biomarkers that flag a potential dysregulated immune response was systematically put together (Material S1) and profiled using the Ella Simple Plex Immunoassay (San Jose, CA, USA). The biomarkers used were as follows: Intercellular Adhesion Molecule 1 (ICAM-1), Lipocalin-2 (LIPO), Myeloperoxidase (MPO), Vascular Cell Adhesion Molecule 1 (VCAM-1), D-Dimer, E-selectin (E-SEL), Ferritin, Surfactant Protein (SP-D), Programmed Death-Ligand 1 (PDL1), Granulocyte Colony-Stimulating Factor (G-CSF), Interleukin 1 beta (IL-1b),

Vascular Endothelial Growth Factor C (VEGFC), Angiopoietin 2 (ANG2), $C - X - C$ Motif Chemokine Ligand 10 (CXCL10), Granulocyte Macrophage Colony-Stimulating Factor (GM-CSF), Interleukin 10 (IL-10), Interleukin 17 (IL-17A), Interleukin 1 receptor antagonist (IL-1ra), Interleukin 6 (IL-6), Interleukin 7 (IL-7), C-C Chemokine Ligand 2 (CCL2), Granzyme B (GRANB), Interferon gamma (IFNg), Interleukin 12 (IL-12), Interleukin 15 (IL-15), Interleukin 2 (IL-2), Interleukin 4 (IL-4), and Tumor Necrosis Factor-alpha (TNF-$\alpha$). The train/test dataset is available in Material S2.

## Validation dataset

A second clinical cohort of ICU COVID-19 patients was considered for validating the model. This cohort took place in Dublin, Ireland. Patients ($n = 50$) were recruited from September 2020 to March 2021. The patients were binarized to 30 recovered and 20 deceased. We have opted to use these patients to validate the model because all patients were critically admitted to the ICU. This cohort was composed of the same 28 biomarkers of the training/testing cohort that were quantified through the same Simple Plex Immunoassay (San Jose, CA, USA). The patients were sampled during their ICU admission (1st day), day three, and 14 days later. The concentration of the available samples was averaged so each patient had one unique value per biomarker. The validation dataset is available in Material S3.

## Dimensionality reduction

The two clinical cohorts employed in this study have concentration levels of 28 biomarkers. It has been reported that a dataset with many dimensions will eventually decrease the performance of a given machine learning algorithm or might produce biased results as the fitting curve is too complex (Taylor, 2019). To reduce the number of variables to characterize a patient, a principal component analysis (PCA) was achieved through the *prcomp* function implemented in the R stats package (version 4.1.2). The loading scores of the principal components 1 and 2 were obtained in the $ rotation element that a *prcomp* object has. The R code (R version 4.1.2) used for reducing the dimensions of the cohorts of this study is found at http://dx.doi.org/10.5281/zenodo.6643238.

## Classification

We have benchmarked five algorithms for classifying the patients from the train/test and validation datasets according to their labels (*i.e.*, 0 deceased and 1 recovered, included in Material S2 and Material S3): support vector machines (SVMs), random forest (RF), classification and regression trees (CART), k-nearest neighbors (KNN), XGBoost, and ANNs. The feasibility of the classification of each algorithm was measured through the accuracy metric upon a 5-fold cross validation dataset. The SVM, RF, CART, and KNN were built in R and the script is available at https://github.com/gustavsganzerla/covid-biomarker/blob/main/different_classif-models.R. The XGBoost classifier was built in Python with the XGBClassifier library (version 1.4.0) and is available at https://github.com/gustavsganzerla/covid-biomarker/blob/main/xgboost_biomarkers_COVID.ipynb.

A deep learning instance of ANNs was used. The ANN simulations took place in the Python language through the TensorFlow library in its version 2.8.0. To scale the

entry data under the same magnitude, a normalization process was conducted through the *StandardScaler* function found in the sklearn.preprocessing library. To validate the simulation process, a cross-validation process was considered, where $k =5$. This method involves reserving 4/5 of the dataset for the training process and the remaining 1/5 for testing. Then, the process is repeated $k$ times so every data point is covered both in training and testing. The training data consists of 80% of the data.

The ANNs were trained, tested, and validated (the latter with external data to grant generalization capacity to the model). To assess a binary prediction, a confusion matrix was built. Deceased patients have the 0 label while recovered patients have 1. This representation of the classification assigns a prediction to a: (i) True Positive (TP), *i.e.*, patients who survived and the model classified as survivors, (ii) True Negative (TN), *i.e.*, patients who deceased and the model indicated so, (iii) False Positive (FP), patients who actually died and the model indicated they survived, and iv) False Negative (FN), *i.e.*, patients who survived and the model indicated they died. The assessment of the model involves calculating the following performance metrics.

Accuracy, which measures the number of correct predictions in the whole dataset; its calculation is achieved by Eq. (1).

$$Accuracy = \frac{(TP + TN)}{(TP + TN + FP + FN)}.$$ (1)

Precision, which measures the ratio between the TPs among all the positives, its calculation is achieved by Eq. (2).

$$Precision = \frac{TP}{(TP + FP)}.$$ (2)

Recall, which measures the percentage of the model identifying TPs; Eq. (3) calculates it.

$$Recall = \frac{TP}{(TP + FN)}.$$ (3)

Finally, the specificity metric calculates the detection rate of TNs throughout the entire dataset. It is obtained through Eq. (4).

$$Specificity = \frac{TN}{(TN + FP)}$$ (4)

Next, the model had its recall and specificity scores assessed to obtain the Receiver Operator Characteristic (ROC) curves, which measured the trade-off between recall and specificity following different thresholds applied to the outcome of the sigmoidal function used in the output neuron. The default threshold value (0.5) was preserved in all instances of the ANN simulation. A second instance of the classification procedure was achieved by isolating the sex of the patients. The Python code that implemented all the ANNs simulations in this study is available at https://github.com/gustavsganzerla/covid-biomarker.

### Ethics approval

The training/testing cohort obtained an Institutional Review Board (IRB) Approval from MetroHealth Medical Center in Cleveland, Ohio IRB 20-00198 on March 25, 2020.

**A**

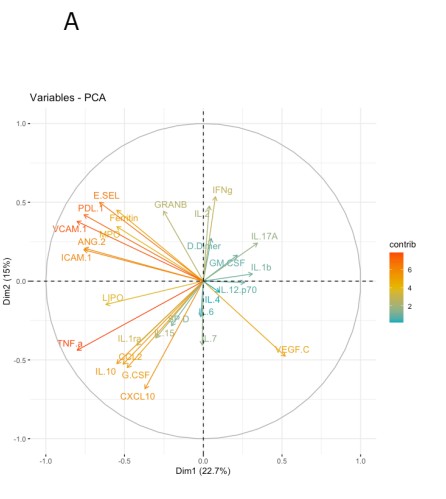

Variables - PCA

**B**

Principal Component 1

| Biomarker | Loading score |
|---|---|
| VCAM.1 | 0.318 |
| TNF-α | 0.317 |
| ICAM.1 | 0.301 |
| PDL.1 | 0.300 |
| ANG 0.2 | 0.298 |
| E.SEL | 0.261 |
| LIPO | 0.245 |
| IL.10 | 0.217 |
| Ferritin | 0.217 |
| MPO | 0.217 |
| VEGF.C | 0.206 |
| CCL2 | 0.202 |
| G.CSF | 0.191 |
| IL.1ra | 0.167 |
| CXCL10 | 0.146 |
| IL.17 | 0.136 |
| IL.1b | 0.123 |
| IL.15 | 0.119 |
| IL.12 | 0.104 |
| GRANB | 0.099 |
| GM.CSF | 0.085 |
| SP.D | 0.080 |
| IL.4 | 0.040 |
| IFNg | 0.030 |
| D.Dimer | 0.020 |
| IL.2 | 0.015 |
| IL.6 | 0.006 |
| IL.7 | 0.002 |

Principal Component 2

| Biomarker | Loading score |
|---|---|
| CXCL10 | 0.333 |
| G.CSF | 0.267 |
| IFNg | 0.261 |
| CCL2 | 0.257 |
| IL.10 | 0.255 |
| E.SEL | 0.243 |
| IL.2 | 0.232 |
| VEGF.C | 0.232 |
| Ferritin | 0.219 |
| GRANB | 0.216 |
| TNF-α | 0.213 |
| PDL.1 | 0.205 |
| IL.1r | 0.198 |
| IL.7 | 0.197 |
| VCAM.1 | 0.184 |
| IL.15 | 0.175 |
| MPO | 0.168 |
| SP.D | 0.138 |
| D.Dimer | 0.131 |
| IL.17 | 0.117 |
| IL.6 | 0.110 |
| ANG 0.2 | 0.101 |
| ICAM.1 | 0.096 |
| GM.CSF | 0.080 |
| LIPO | 0.072 |
| IL.4 | 0.034 |
| IL.1b | 0.022 |
| IL.12 | 0.004 |

**Figure 1   Principal component analysis to reduce the dimensionality in the train/test cohort¡.** In (A), we plot the contribution of each variable to the Principal Component (PC) 1 and PC2, (these are responsible for 21.5% and 14.7% of the variance of PC1 and PC2, respectively). In (B), we targeted PC1 and PC2 and their eigenvalues. Then, we decreasingly sorted the loading scores of these two components and extracted the name of each variable. Therefore, the biomarkers indicated in (B) are–in decrescent order–the proportion that each variable has in computing the variance of PC1, and PC2.

The validation cohort obtained an IRB Approval from SJH/TUH Joint Research Ethics Committee and The Health Research Consent Declaration Committee (HRCDC) under the register REC: 2020-05 List 17 on March 2, 2020.

# RESULTS

## Dimensionality reduction

To reduce the dimensionality of the dataset and consequently have fewer neurons being used in the input layer, we applied a Principal Component Analysis (PCA). From it, we selected the first two principal components (PCs) (Fig. 1A); then we measured the loading score in these two components and extracted the variables that most contributed to explain the components' variance (Fig. 1B). PC1 is responsible for 22.7% of the data variance while PC2 is responsible for15%. We maintained this proportion and selected three biomarkers from PC1 and two from PC2. Therefore, we report a reduced number of biomarkers that explains the variability found in the dataset; *i.e.*, VCAM.1, TNF- $\alpha$, ICAM.1, CXCL10, G.CSF. In addition, we have also screened the 5 worst-ranking biomarkers by the PCA for validation purposes (we maintained the same proportion as that used in the 5 best-ranking PCA biomarkers).

## Defining the classification procedure

We selected RF, SVMs, CART, XGBoost, KNN, and ANNs to classify the data. As the accuracy performance of the first four techniques did not show satisfactory results in

correctly labeling the train/test and validation cohorts (Material S4), we therefore opted to use Deep Learning ANNs as the previous models did not show a capacity of generalizing upon unseen data.

To determine the optimal architecture of ANNs to be used in this work, a series of parameters were set. First, the input layer of the ANN contains the number of biomarkers that will be employed to classify patients' outcomes and nature of infection. Next, we created different ANN models to determine the optimal number of hidden layers and their number of neurons (Material S5). From that, we opted to use an architecture consisting of four hidden layers with 20 neurons each, where the neurons are activated by a ReLU function. The loss function was set to binary cross entropy and the Adam optimizer was chosen. We allowed the weights of the ANN to be updated 200 epochs; this was found to be the limit in which the error did not drop any further. Since all the further prediction steps deal with binary classification, we set one neuron in the output layer; this neuron is activated by a sigmoid function since it yields a probability (*i.e.*, probability of recovering or dying).

### Classifying patients' outcome with 28 biomarkers

We predicted the clinical outcome of COVID-19 patients using 28 biomarkers as the input of the ANN model. The results (Fig. 2A) suggest a satisfactory classification of the patients. We further explored our classification metrics by providing the ROC curve (Fig. 2B) in which we adjusted the decision threshold obtained as the output by the sigmoidal function in the output neuron. Lastly, we show in Fig. 2C, the error dropping with increasing epochs.

### An optimal set of biomarkers succeeds in classifying patients' outcomes

To test the prediction capacity of the biomarkers identified by the PCA, we ran an ANN with five neurons in the input layer. To validate the prediction power of the ANN with the PCA biomarkers, we performed two classification procedures with different input sets: one with the five best-ranking PCA biomarkers (three from PC1 and two from PC2) in the input layer and a second with the five worst-ranking PCA biomarkers (three from PC1 and two from PC2). In Fig. 3, we show the distinctive classification obtained by both sets of biomarkers (Fig. 3A). Next, we show the ROC curves for each point in the decision threshold (Fig. 3B). Finally, in Fig. 3C, we show the error drop rate, during the execution of both ANNs. We found the ANN trained with the 5-best ranking PCA biomarkers ANN dropped its error at a significantly different rate than its counterpart ($p = 2.2e{-}16$) with the increase of epochs. In addition, due to the time of enrollment, the patients from the train/test cohort were likely infected with the Wuhan D614G strain of the virus.

We combined male and female patients of the two cohorts in separate datasets to assess the classification capacity gender-wise (Table 1). We maintained the same input biomarkers identified by the PCA analysis. We found that the classification of female patients outperforms the metrics of male patients despite the accuracy of male patients being higher than that of females (*i.e.*, 77% *vs.* 73.66%, respectively).

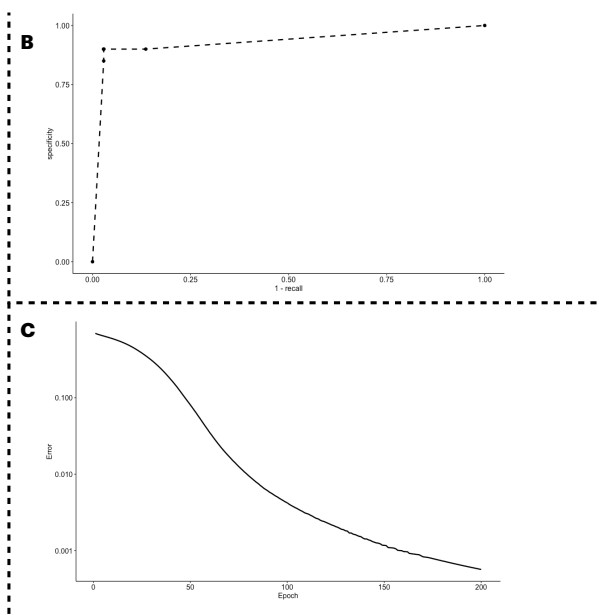

| | |
|---|---|
| Accuracy | 94% |
| Precision | 93.2% |
| Recall | 97% |
| Specificity | 90% |

**Figure 2  ANN predicting the patients' outcome of the train/test cohort by using all biomarkers information available.** We show the performance metrics for the ANN trained with all biomarkers as input in (A), the numbers displayed here were obtained with the decision threshold set to 0.5 on the sigmoidal function implemented in the output neuron, *i.e.*, a given patient is labeled as 1 if they have recovered, otherwise, the patient is assigned with a 0. In (B), we show the variation of the decision threshold from 0 to 1 (increasing every 0.1th interval) in the classificatory task. In (C), we show the error rate, obtained with the binary cross entropy function implementation, the error was averaged in each instance of the *k* cross validation process, totalling 5 times.

**Table 1  Classification with patients' sex isolated.** The patients from the train/test and validation cohorts were combined, then, we isolated the female and male patients of each dataset. A classification procedure through artificial neural networks took place by preserving the architecture used previously. Also, the input of this classification is the biomarkers isolated by the principal component analysis step. Performance metrics were recorded after 200 learning epochs and the threshold set in the output of the sigmoid function neuron was set to 0.5.

| | Male | Female |
|---|---|---|
| Accuracy (%) | 77 | 73.66 |
| Precision (%) | 70 | 83.11 |
| Recall (%) | 75.55 | 82.33 |
| Specificity (%) | 83.50 | 87.5 |

## The prediction capacity of individual biomarkers

To test the clinical outcome prediction capacity of each biomarker, we ran individual ANNs with a single neuron as input. In these regards, we observed the ROC curves for each individual biomarker (Fig. 4) and compared it to how the error dropped after 200 epochs in the ANN model (Fig. 5). The same biomarkers with satisfactory ROC scores matched the low error after 200 epochs (Table 2).

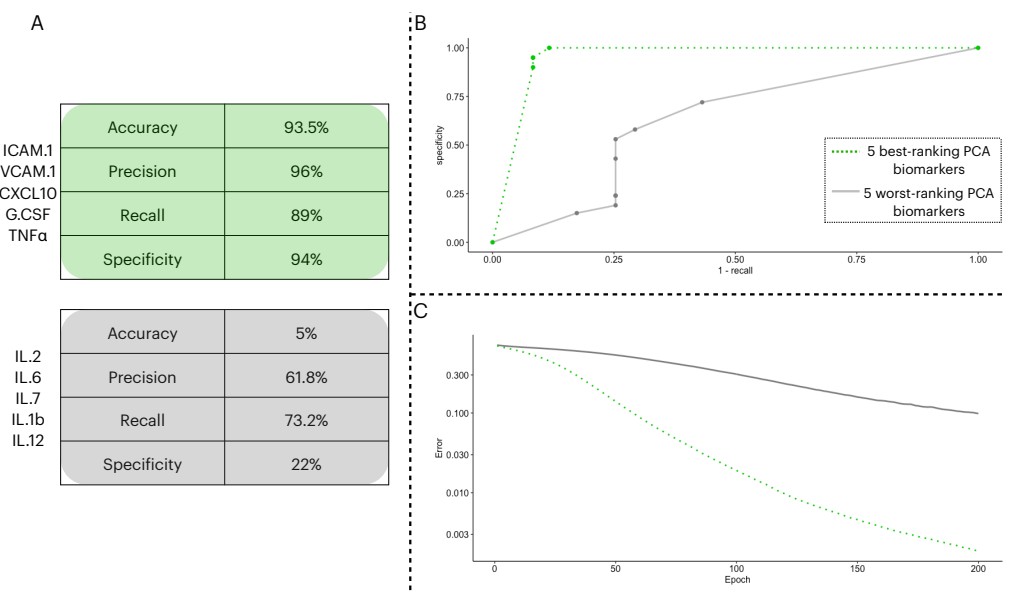

**Figure 3  Two ANN simulations seeking to predict COVID-19 patients outcome based on different sets of biomarkers.** In (A), we show the performance metrics for one ANN trained with the top-five biomarkers (green) and another ANN with the least-five biomarkers (gray) from the PCA implemented in 3.3. The values presented are sigmoidal function output in its threshold 0.5 of the averaged cross validated testing dataset. In (B), we depict the ROC curve for the two ANNs ranging from 0 to 1 in the decision threshold. In (C), we show how the error (log 10 $y$ axis) decreased with increasing epochs ( $x$ axis), the error was averaged in each instance of the $k$ cross validation process, totalling five times.

The biomarkers that individually showed a satisfactory prediction capacity (balanced performance metrics) are TNF-$\alpha$, PDL1, LIPO, ICAM1, and VCAM; from these, TNF-$\alpha$, ICAM.1, and VCAM are the main contributors of the PC1 found by the PCA analysis.

## Validation of the findings with an external cohort

In order to provide validation to the models previously identified, an external source of COVID-19 patients. At that time, the circling variant in the British Isles was B.1.1.7. In Fig. 6A, we assess the generalization capacity of three different classification approaches explored in this work. From it, the patients were divided into two classes (*i.e.*, recovered and deceased), and each class was validated with the classification models. Firstly, the classification with 28 biomarkers only classified the deceased patients correctly. In the second approach, the five best ranking PCA biomarkers produced a balanced prediction of the two classes. Finally, the classification with the best individual biomarker (*i.e.*, TNF-$\alpha$) also only managed to yield a satisfactory classification of deceased patients. In Fig. 6B, we isolate the model that well generalized the external data (*i.e.*, five best-ranking PCA biomarkers) and map the validation data into the ANN's sigmoid function for which both deceased and recovered patients were satisfactorily classified. The average function outcome is 0.42 and 0.61 for deceased and recovered, respectively.

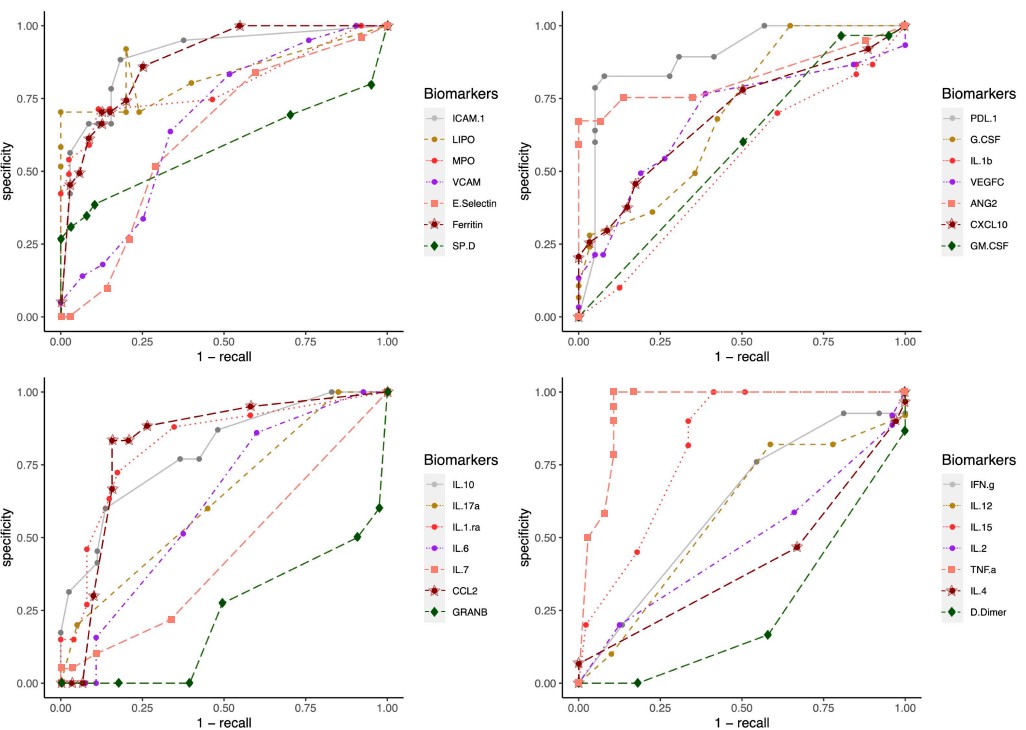

**Figure 4   ROC curves for individual biomarkers.** ROC curves were plotted for the prediction capacity of each biomarker present in the study. The decision threshold was adjusted ranging from 0 to 1 (increasing every 0.1th interval).

## DISCUSSION

In this work, we promote a two-sided interpretation of a biomarker analysis, one regarding the potential use of biomarkers for future streamline diagnosis tests and a second one revolving around the representation of a dysregulated immune response reflected by unbalanced biomarkers. The results we obtained enabled the isolation of five immune biomarkers namely ICAM.1, VCAM.1, G.CSF, CXCL10, and TNF-$\alpha$. The differentiated concentration of immune biomarkers is known to play a key role in regulating host response against pathogens (*Yang et al., 2014*; *Bermejo-Martin et al., 2020*; *Bowman et al., 2021*). We could employ these flagship proteins as an input of ANNs to separate COVID-19 patients based on their outcomes (*i.e.*, recovered and deceased).

When we first used SVM, XGBoost, KNN, RF, and CART to classify our data, we noticed the models obtained did not show a capacity of generalization since they did not produce balanced accuracy scores in the two clinical cohorts. We then chose a more robust classifier, *i.e.*, ANN. We show its success by using three different sets of biomarkers: (i) all the 28 available, (ii) the five best-ranking PCA biomarkers (we also used the five worst-ranking PCA biomarkers), and (iii) the individual biomarker that better showed classification capacity, *i.e.*, TNF-$\alpha$. Within the train/test data, all sets of biomarkers yielded in a satisfactory discriminant model. However, when we stressed the 28-biomarker and

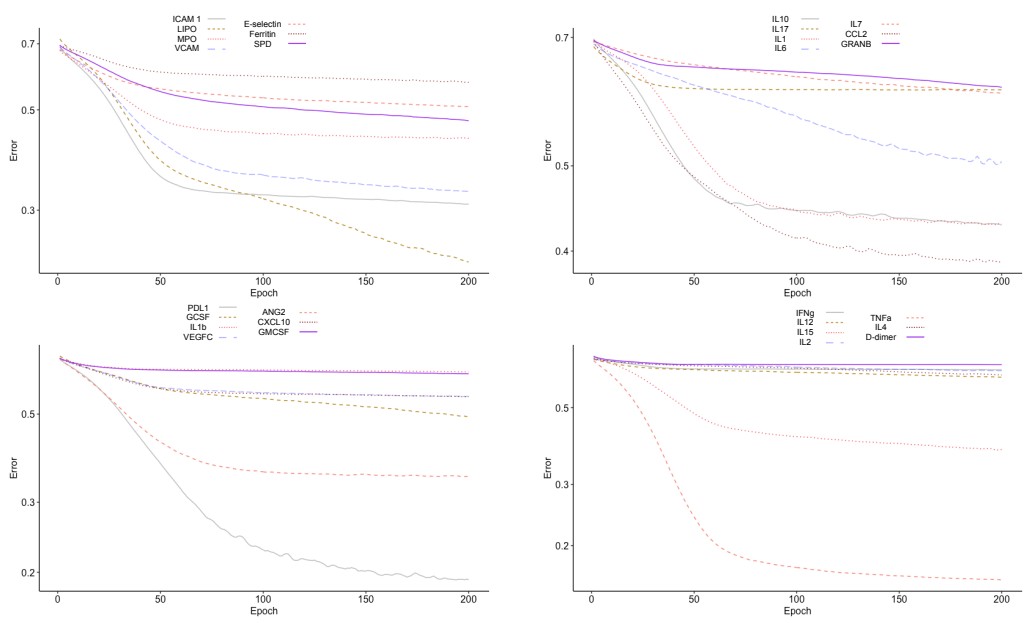

**Figure 5 Error drop rate for individual biomarkers.** The error returned by the binary cross entropy function decreases providing that more iterations over the training dataset occur.

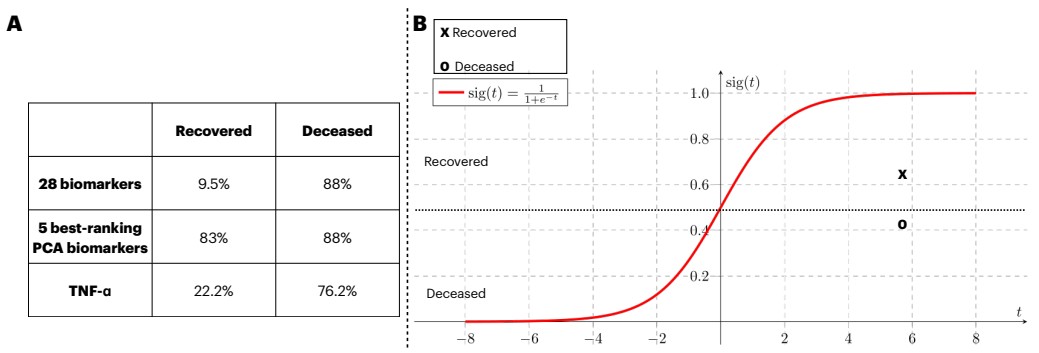

**Figure 6 External validation of three ANN-based classification rationales.** We have performed a validation process with data not included in the train/test of the ANN. The three classification rationales explored in this study (*i.e.*, all biomarkers, the best PCA biomarkers, and TNF-$\alpha$). The validation data was sliced in two according to the clinical outcome (deceased and recovered) and the results of the classification are displayed in (A). In (B), we mapped the average outcome value for the two slices of the validation dataset into the sigmoid function. The same threshold achieved in the train/test set was preserved, so the function was divided into the ranges 0 to 0.49 and 0.5 to 1 for deceased and recovered patients, respectively.

the TNF-$\alpha$ models with an external set of patients, the ANNs failed to correctly classify the recovered patients. Notwithstanding, the five best-ranking PCA biomarkers model was successfully validated, granting a generalization capacity to our model. In fact, the ability to generalize upon other data has always been a daunting challenge since the early conceptualization of ML techniques (*Leung & Chow, 1999*). Next, to determine the five

**Table 2 Individual predictive capacity of 28 biomarkers.** Individual ANN simulations took place with a single neuron in the input layer. The remaining ANN architecture was preserved, matching the other instances employed in this study. The measurements displayed in Table 1 represent the ANN on its 200th learning epoch.

| | Error drop rate | Accuracy | Precision | Recall | Specificity |
|---|---|---|---|---|---|
| TNF-$\alpha$ | 0.1591 | 91.77 | 97.14 | 89.28 | 95 |
| PDL.1 | 0.19144 | 89.55 | 90.55 | 92.14 | 82.66 |
| LIPO | 0.23034 | 89.77 | 86.42 | 99.9 | 70.33 |
| ICAM.1 | 0.30968 | 75.77 | 79.76 | 84.57 | 66.33 |
| VCAM | 0.3299 | 79.55 | 82.66 | 87.42 | 70.33 |
| ANG.2 | 0.34872 | 81.33 | 79.04 | 93.14 | 67.33 |
| IL.15 | 0.37874 | 73.33 | 80.33 | 66.44 | 81.66 |
| CCL2 | 0.38888 | 75.33 | 78.85 | 84.28 | 66.66 |
| IL.10 | 0.4282 | 73.33 | 76.66 | 86.3 | 60 |
| IL.1ra | 0.42956 | 74.88 | 73.92 | 85.14 | 63.33 |
| MPO | 0.4324 | 77.77 | 80.59 | 88.45 | 71.33 |
| SP. D | 0.47296 | 67.11 | 68.25 | 92.14 | 34.66 |
| G.CSF | 0.49266 | 61.11 | 66.78 | 77.42 | 36 |
| IL.6 | 0.50564 | 59.33 | 63.01 | 89.16 | 15.66 |
| E. Selectin | 0.50846 | 59.55 | 62.95 | 74.76 | 33.66 |
| CXCL10 | 0.55304 | 65.55 | 67.65 | 85.11 | 37.66 |
| VEGFC | 0.55376 | 62.88 | 71.5 | 81 | 49.33 |
| Ferritin | 0.57496 | 56.88 | 63.55 | 79.14 | 27 |
| IL.7 | 0.60488 | 59.11 | 60.85 | 89.33 | 10 |
| IL.17 | 0.61002 | 60.88 | 64.88 | 95 | 20 |
| IL.12 | 0.61236 | 55.33 | 61.33 | 90 | 10 |
| GRANB | 0.6142 | 28.44 | 47.68 | 60.5 | 0 |
| IL.4 | 0.62172 | 66.33 | 62.88 | 99.9 | 6.6 |
| GM.CSF | 0.63154 | 60.88 | 60.88 | 99.9 | 0 |
| IL.1b | 0.63826 | 53.77 | 60.77 | 87.5 | 10 |
| IL.2 | 0.63954 | 55.33 | 65.33 | 87.5 | 20 |
| IFN.g | 0.64308 | 51.11 | 63.11 | 45.52 | 76 |
| D-Dimer | 0.66516 | 48.66 | 54.66 | 81.9 | 0 |

best-ranking PCA biomarkers as potential candidates for isolation, we compared the classification capacity brought by them against the five worst-ranking PCA biomarkers. The results advocate for the five biomarkers to be a good representative of the data variance captured by the PCA (*Gárate-Escamila, Hajjam El Hassani & Andrès, 2020*). Finally, by isolating the patients from the two cohorts by gender, we found that a classification only with females yielded more satisfactory results.

In terms of their function, the five biomarkers we isolated with the PCA, *i.e.*, ICAM.1, VCAM.1, G.CSF, CXCL10, and TNF-$\alpha$, are cytokines/chemokines that mark inflammatory responses. Such inflammation, in COVID-19, can develop into systemic inflammation and consequently, fatality (*Hartung, 1998*; *Mukhopadhyay, Hoidal & Mukherjee, 2006*;

*Vazirinejad et al., 2014*; *Müller, 2019*; *Zhang et al., 2020*). The mutual functioning of VCAM.1 and ICAM.1 have been reported as markers of inflammation in patients with slow coronary flow (*Turhan et al., 2006*). Furthermore, CXCL10, TNF-$\alpha$, and G.CSF have been found to be positively correlated with morbidity in past respiratory pandemics (*McHugh et al., 2013*). Interestingly, the ANN model was able to capture the unbalanced levels of these biomarkers and use the information to predict whether one recovers or not.

Moreover, our individual biomarkers model succeeded in classifying deceased patients based on their TNF-$\alpha$ concentration. This biomarker has been observed to disturb the balance of signaling complexes, potentially resulting in inflammatory cascades (*Webster & Vucic, 2020*), and evidence of its being linked with death prediction has also been reported (*Bruunsgaard et al., 2003*). ANNs usually function as a black-box classifier, in which extracting real-world information about the behavior of its variables is not a practical task. However, with the results we obtained regarding TNF-$\alpha$, we suggest that off-concentrations of it might be used to depict the unbalanced immune response of COVID-19 patients.

Machine learning (ML) techniques can solve problems in areas where mechanistic statistics are not able to. In fact, the mathematical robustness implemented by instances of ML can explore the complex relationship of several variables in a non-linear way (*Bühlmann, 2020*). There have also been efforts to predict the clinical outcome of COVID-19 patients by utilizing simpler (than ANNs) ML approaches, such as logistic regression (*Arnold et al., 2021*) and support vector machines (*de Souza et al., 2021*). Both these techniques yielded satisfactory two-class prediction. However, their nature is quite different from the one encompassed in this study. First, (*Arnold et al., 2021*) brought biomarkers collected only on admission to their classification rationale, moreover *de Souza et al. (2021)* have employed demographical data to achieve their classification, highlighting the differential analysis protruded by the inclusion of immune biomarkers.

We also noticed the inability of two (out of three) ANN models to correctly predict recovered patients from a validation cohort. All the patients in this cohort were recruited in the ICU; also, due to limitations in binarizing these patients into two classes (*i.e.*, recovered and deceased, as found in the training/testing cohort), the recovered label in this cohort means the patients were discharged from ICU but remained hospitalized. Therefore, we argue that inflammatory activity was still happening within those patients, which is why more generic models (*i.e.*, classification with all biomarkers and classification with TNF-$\alpha$) failed to correctly label those patients.

A reduced set of immune biomarkers identified by this work has succeeded in predicting the outcome of patients. In fact, in frameworks for therapeutic development (*CDER, 2014*), a set of candidate biomarkers is identified for further tests to validate the indirect/direct causal relation between a biomarker, a disease, and its treatment (*Kraus, 2018*).

We faced limitations in this study due to the reduced size obtained in the cohorts. To overcome this issue and grant validity to our findings, we opted to train/test the model with one dataset and use an additional set for validation. In addition, other studies that used biomarkers from clinical cohorts (*Bermejo-Martin et al., 2020*; *Fazolo et al., 2021*; *Sardar, Sharma & Gupta, 2021*) did not show a substantial improvement in their sample population to ours. Another potential limitation we faced was due to the classificatory

nature of this study, which asks for categorical variables (*i.e.*, recovered or deceased) to classify patients. For that, regression models (*i.e.*, Recurrent Neural Networks) can analyze how numerical variables behave over time.

Therefore, we suggest that an extensive round of tests, such as the one achieved by the employment of deep learning ANNs in this work, functions as a form to curate biological information in an *in-silico* way as the non-linear relationship among a plethora of biomarkers might not be explored by conventional statistical approaches. We believe that our results might be an initial step for feeding models for drug targeting, highlighting *in-silico* biology as an economical way to tailor hypotheses to be further investigated by molecular and experimental analyses.

## CONCLUSIONS

In this work, we have identified a neural network architecture and stressed this classification model with different inputs to achieve an optimal classification. In all instances, the training/testing dataset was satisfactorily predicted by the model we proposed. Next, when validated with external data, we were able to screen one subset of biomarkers that correctly predicted the clinical outcomes of COVID-19 patients belonging to two cohort studies.

## ACKNOWLEDGEMENTS

We thank the Canadian Research Chair in Translational Vaccinology and Inflammation. We thank Michelle Hecker and Claudia Ute Sontich from MetroHealth Medical Center and Banumathi Tamilselvan from Case Western Reserve University for their contributions in patient coordination and sample acquisition. We thank all the members from the Laboratory of Emerging and Infectious Diseases at Dalhousie University. Finally, we also thank Nikki Kelvin for her key inputs to this manuscript.

### Funding

This work was supported by awards from the Canadian Institutes of Health Research, the Canadian 2019 Novel Coronavirus (COVID-19) Rapid Research Funding initiative (CIHR OV2 –170357), Research Nova Scotia (David Kelvin), Atlantic Genome/Genome Canada (David Kelvin), Li-Ka Shing Foundation (David Kelvin), and the Dalhousie Medical Research Foundation (David Kelvin). This study has also been supported by SFI (Science Foundation Ireland), Grant Number 20/COV/0038 (IML). This work was also supported by a donation from the Nord Family Foundation to Mark Cameron, Ann Avery, and Cheryl Cameron. The funders had no role in study design, data collection and analysis, decision to publish, or preparation of the manuscript.

### Grant Disclosures

The following grant information was disclosed by the authors:
Canadian Institutes of Health Research.

the Canadian 2019 Novel Coronavirus (COVID-19).
Rapid Research Funding initiative: CIHR OV2 –170357.
Research Nova Scotia.
Atlantic Genome/Genome Canada.
Li-Ka Shing Foundation.
Dalhousie Medical Research Foundation..
Science Foundation Ireland: 20/COV/0038 (IML).
Nord Family Foundation.

## Competing Interests

The authors declare there are no competing interests.

## Author Contributions

- Gustavo Martinez conceived and designed the experiments, performed the experiments, analyzed the data, prepared figures and/or tables, authored or reviewed drafts of the article, and approved the final draft.
- Alexis Garduno conceived and designed the experiments, performed the experiments, analyzed the data, authored or reviewed drafts of the article, and approved the final draft.
- Abdullah Mahmud-Al-Rafat conceived and designed the experiments, performed the experiments, analyzed the data, authored or reviewed drafts of the article, and approved the final draft.
- Ali Toloue Ostadgavahi conceived and designed the experiments, performed the experiments, analyzed the data, authored or reviewed drafts of the article, and approved the final draft.
- Ann Avery conceived and designed the experiments, performed the experiments, analyzed the data, authored or reviewed drafts of the article, and approved the final draft.
- Scheila de Avila e Silva conceived and designed the experiments, performed the experiments, analyzed the data, authored or reviewed drafts of the article, and approved the final draft.
- Rachael Cusack conceived and designed the experiments, performed the experiments, analyzed the data, authored or reviewed drafts of the article, and approved the final draft.
- Cheryl Cameron conceived and designed the experiments, analyzed the data, authored or reviewed drafts of the article, and approved the final draft.
- Mark Cameron conceived and designed the experiments, analyzed the data, authored or reviewed drafts of the article, and approved the final draft.
- Ignacio Martin-Loeches conceived and designed the experiments, analyzed the data, authored or reviewed drafts of the article, and approved the final draft.
- David Kelvin conceived and designed the experiments, analyzed the data, authored or reviewed drafts of the article, and approved the final draft.

## Human Ethics

The following information was supplied relating to ethical approvals ({i.e.}, approving body and any reference numbers):

The training/testing cohort obtained an Institutional Review Board (IRB) Approval from MetroHealth Medical Center in Cleveland, Ohio IRB 20-00198 on March 25th, 2020. The validation cohort obtained an IRB Approval from SJH/TUH Joint Research Ethics Committee and The Health Research Consent Declaration Committee (HRCDC) under the register REC: 2020-05 List 17 on March 2nd, 2020.

## Ethics

The following information was supplied relating to ethical approvals ({*i.e.*}, approving body and any reference numbers):

MetroHealth Medical Center;

SJH/TUH Joint Research Ethics Committee and The Health Research Consent Declaration Committee.

## Data Availability

All the codes used for deriving the conclusions of this study is available at GitHub: https://github.com/gustavsganzerla/covid-biomarker.

The input datasets for the machine learning procedures are also available at Supplementary Files.

## Supplemental Information

Supplemental information for this article can be found online at http://dx.doi.org/10.7717/peerj.14487#supplemental-information.

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
