# Peer review of "An artificial neural network classification method employing longitudinally monitored immune biomarkers to predict the clinical outcome of critically ill COVID-19 patients"

_PeerJ, doi:10.7717/peerj.14487_

## Round 0.1 · original submission · Major Revisions

Your manuscript has been reviewed and requires several modifications before making a decision. The comments of the reviewers are included at the bottom of this letter. Especially, Reviewers 2, 3, and 5 indicated that the introduction and methods sections should be improved. I agree with the evaluation, and I would request the manuscript be revised accordingly.

Reviewer 1 ·

Basic reporting

I think this is a excellent research using deep learning neural network models to explore the relationship that exists between immune biomarkers and clinical outcomes in critically ill Covid-19 patients.
Basic comments as follow:
1. Clear and unambiguous, professional English used throughout.
2. Sufficient field background/context provided.
3. Professional article structure, figures, tables. Raw data shared.
4. This study provides logical and rigorous analysis and reasoning of the results, demonstrating that appropriate evaluation metrics support the performance of the findings.

Experimental design

Comments:
1. The research question is well-defined, relevant, and meaningful.
2. The research provides an ethical table and ensures that rigorous investigations are conducted with high technical and ethical standards.
3. Some neural network details can be considered to be added to ensure the reproducibility of the experiments. For example, the loss function (cross entropy or other) used by the neural network, the choice of the optimizer during the optimization of the model parameters.
4. In addition, the author can also consider trying to adjust the number of network layers and neural units of the neural network model, compare them, and then select the model with the best performance.

Validity of the findings

Appropriate evaluation metrics were used in this study to validate the performance of the model, and rigorous cross-validation was performed using real clinical data. So I think the results of this study are reliable.
All underlying data have been provided; they are robust, statistically sound, & controlled.
Conclusions are well stated, linked to original research question & limited to supporting results.

Additional comments

I'm impressed with the research, but there are still some gaps. The data involved in this study are more based on time sequences. For example, patients may have different immune biomarkers at different stages of disease. Therefore, when choosing a deep learning algorithm, researchers should consider whether the algorithm can correctly express and interpret time sequences. Future research can consider using recurrent neural networks to analyze and discuss the data in this study (with time sequences) in depth at different time steps.

Reviewer 2 ·

Basic reporting

The article provides an interesting attempt to use an artificial neural network to predict a binary Covid-19 outcome, recovered or deceased, based on a set of biomarkers. The study as a whole provides a significant and worthy endeavor but the manuscript needs more work.
The abstract needs shortening. I recommend removing the first two sentences (lines 38-39)- they are better in the introduction. The first sentence in the results in confusing and unnecessary (line 50). However, the outcomes tested should be specified in line 42, as the generic "outcomes" can mean different things than tested here.
In terms of the report, the introduction needs the most work and the manuscript gets better as it progresses.
The introduction needs more theoretical depth. Why are these specific biomarkers chosen? What is the predictive use besides just predicting death? There is some mention of the latter but the argument is not compelling. It is addressed to a certain degree in the discussion but there is little or no introduction leaving the reader with the feeling this is all about predicting death. If it is to what purpose? I hope it’s not withholding medical treatment. I believe this isn’t the argument, but the case for what the useful value of such prediction is needs to be made stronger.
In the ANN section of the method, it is not stated what outcome is considered “positive” and which “negative”
The term “recall” for the proportion of correct positives compared to all true positives, is not intuitively obvious. Unlike the other labels, it is not clear why this label is used and an explanation should be given or a different label used.
The reference to sections 3.2, 3.4 etc. is not clear- state where these can be found. (line 186 and later)
The graphs and tables are useful but I would strongly recommend to identify data with more than colors. This is important both in terms of possible color deficiencies of the reader and possible different reading platforms- different types of lines, like the ones used in Figure 4, for example, are very useful, but the line type should be included in the legend (not just the colored dot)

Minor issues:
Line 74- "it is required" should be removed the sentence should start " an extensive approach is required..."
Line 75- "of tool" should be "of a tool"
Line 77- sentence starting with "The classificatory instance" needs revising. the wording is unclear and cumbersome.

Experimental design

The exclusion of patients outside of the confidence interval needs more justification and some information about them, e.g. Line 98. The argument that their inclusion is “negatively affecting the results” is not an acceptable justification. Usually, outliers are considered such when they are three standard deviations farther than the mean which doesn’t seem to be the case here.

The biomarkers were collected at multiple times and were averaged in the analysis, which may be justifiable although it would be interesting to see the progression of these Biomarkers at different times. However, the claim that they were “averaged to encompass the evolution of the patients” is confusing (line 128). This statement suggests an analysis of how the biomarkers change over time, which is not the case. An explanation of why this was not attempted is also warranted in this section (or maybe in future research directions, see Validity of Findings section).

Validity of the findings

This study provides an interesting used of Neural Networks to predict death or recovery in the short term but the importance of such predictions is not analyzed enough theoretically. There is some reference to the meaning of these finding in the discussion. More extensive discussion of the implications of the predictive markers would make the manuscript stronger. Also, what are the future directions of this work? What more can be done with these predictions? What about predicting outcomes beyond death in a short period of time? Are change patterns in the biomarkers to be examined further? While one study is not expected to address all these issues a discussion of them is warranted to contextualize the work.

Reviewer 3 ·

Basic reporting

No comment.

Experimental design

1. The first important concern is that only ANN is implemented and evaluated, without comparing with some baselines, e.g., XGBoost.

2. The second concern is that the dataset is very small, while tabular-based machine learning models could be better because deep learning models are prone to overfitting.

3. The third concern is that the process of determining the optimal ANN structure is not given, making the findings less convincing.

Validity of the findings

No comment.

Additional comments

Check the statement "The training data consists of 20% of the data." in line 149 Page 9. 20% could be 80%.

Reviewer 5 ·

Basic reporting

The manuscript is well-written with clear English. Sufficient background is provided with professionally structured display items.

Experimental design

It's good that the authors validated their model in an independent patient cohort. However, the following experimental designs are problematic:
1) Use of ANN is not justified in this study. The authors should first show performance comparison with linear classification model and SVM, and use ANN if the less complex models fail to achieve good performance in classification. Especially linear models, they provide good transparency in model prediction and should be the to-go especially in clinical setting if linear models have comparable performance to ANN.
2) Given the known variety of response among COVID-19 patients, the reasoning for removing patients do not seem valid and necessary. Please also show performance when including all patients.
3) For parameter tuning, the model achieving best validation performance should be used. This rule-of-thumb applies to hyper-parameters including number of epochs.

Validity of the findings

The authors should also control for gender, ethnicity, and viral strain whenever possible to ensure generalizability and validity of finding.

Reviewer 6 ·

Basic reporting

English language needs a little improvement.

The introduction is too succinct. There have been several studies on the role of biomarkers on the severity of inflammatory diseases. Pls add a few of them in the introduction section, for the benefit of the readers.

Article structure is good, except that the heading 'Conclusions' is missing in the abstract.

Results are relevant to the hypothesis.

Experimental design

The research is within the aims and scope of the journal.

Research uses Artificial Neural Network/Machine Learning to predict biomarkers that could be involved in worst-case scenarios of COVID-19 patients. The research question is well defined and fills a knowledge gap.

Investigation is rigorous following technical and ethical standards.

Methods are sufficient and clear.

Validity of the findings

Although several biomarkers have been identified as responsible for the severity of inflammatory diseases, the use of ANN/ML makes this study novel.

All data have been provided.

Conclusions are clear.

Additional comments

The authors have conducted an important clinical study on several patients to identify potential biomarkers that can predict the worst-case outcome of COVID-19 patients. For this purpose, they have used machine learning, which I understand is the novelty of this study. I commend the authors for putting in lots of effort in conducting this study. Overall, my impression regarding the study and the manuscript is very positive. However, I have some minor concerns that I would like to see addressed.



Introduction:

Line 67: Use ‘high potential for transmission’ instead of ‘high spreading capacity’
Lines 74 to 75: Pls rephrase the sentence.

Pls check the English language throughout.

Methodology:
Line 142 – The abbreviation ‘ANN’ has already been used earlier.

---

## Round 0.2 · accepted · Accept

Thank you very much for the submission of a revised version of your paper. The reviewers all recommend acceptance and I have gone through the revised, track-changes manuscript and rebuttal letter, and see that the authors addressed the reviewers' concerns and substantially improved the content of the manuscript. So, based on my assessment as an academic editor, the manuscript may be now accepted for publication.

Reviewer 1 ·

Basic reporting

The author further gives detailed data on the machine learning model selection process and the parameter selection process, which satisfies the doubts about this point in my comment. Also, I am impressed that the authors have carefully considered the suggestion of using recurrent neural networks for processing time sequential data.

Experimental design

1. The research question is well-defined, relevant, and meaningful.
2. The research provides an ethical table and ensures that rigorous investigations are conducted with high technical and ethical standards.
3.Data on model selection and parameter selection were added to provide greater rigour and confidence in the experimental design.

Validity of the findings

Appropriate evaluation metrics were used in this study to validate the performance of the model, and rigorous cross-validation was performed using real clinical data. So I think the results of this study are reliable.
All underlying data have been provided; they are robust, statistically sound, & controlled.
Conclusions are well stated, linked to original research question & limited to supporting results.

Reviewer 2 ·

Basic reporting

The article is greatly improved. The language is clear. The theoretical background is sufficient to provide a good rational for conducting this study. The data presentation is now clear, the figures are now accessible to readers with perceptual color deficiency.
minor error- line 448- 449: the citation should not be in parentheses.

Experimental design

The experimental design is adequate and prior concerns regarding it and the analyses have been addressed well.

Validity of the findings

This study provides a meaningful addition to the literature.
While this research has shortcomings, all research does, it provides an important contribution to this area. The discussion now includes important recognition of shortcomings and suggestions for future research.

Additional comments

The authors have clearly paid close attention to the reviewers' points. In my opinion, they have clarified and/or addressed all of them satisfactorily. They certainly addressed all on my concerns.

Reviewer 3 ·

Basic reporting

no comment

Experimental design

no comment

Validity of the findings

no comment

Additional comments

no comment

Reviewer 5 ·

Basic reporting

n/a

Experimental design

n/a

Validity of the findings

n/a

Additional comments

The authors have well-addressed my previous comments.

Reviewer 6 ·

Basic reporting

Good

Experimental design

Good

Validity of the findings

Valid

Additional comments

The authors have addressed all the concerns raised by me. I recommend acceptance of the manuscript.